# Carbon-Supported KCoMoS$_2$ for Alcohol Synthesis from Synthesis Gas

Mohamed E. Osman [1,2], Vladimir V. Maximov [1], Viktor S. Dorokhov [1], Viktor M. Mukhin [3], Tatiana F. Sheshko [2], Patricia J. Kooyman [4,*] and Viktor M. Kogan [1]

[1] N.D. Zelinsky Institute of Organic Chemistry RAS, 119991 Moscow, Russia; wadalmsna3.com@gmail.com (M.E.O.); maximovzioc@gmail.com (V.V.M.); viktor.s.dorokhov@yandex.ru (V.S.D.); kogan@akado.ru (V.M.K.)

[2] Department of Physical and Colloidal Chemistry, Peoples' Friendship University of Russia, 117198 Moscow, Russia; sheshko-tf@rudn.ru

[3] ENPO "Neorganika", JSC, 144001 Electrostal, Russia; viktormukhin@yandex.ru

[4] Chemical Engineering, University of Cape Town, Private Bag X3, Rondebosch 7701, South Africa

[*] Correspondence: patricia.kooyman@uct.ac.za

**Abstract:** KCoMoS$_2$ was supported on various carbon support materials to study the support effect on synthesis gas conversion. Next to two activated carbons with high micropore volume, a traditional alumina ($\gamma$-Al$_2$O$_3$) support and its carbon coated form (CCA) were studied for comparison. Coating alumina with carbon increases the selectivity to alcohols, but the AC-supported catalysts show even higher alcohol selectivities and yields, especially at higher temperatures where the conversions over the AC-supported catalysts increase more than those over the $\gamma$-Al$_2$O$_3$-based catalysts. Increasing acidity leads to decreased CO conversion yield of alcohols. The two activated-carbon-supported catalysts give the highest yield of ethanol at the highest conversion studied, which seems to be due to increased KCoMoS$_2$ stacking and possibly to the presence of micropores and low amount of mesopores.

**Keywords:** syngas conversion; KCoMoS$_2$; carbon support material; supported transition metal sulfide catalysts; ethanol synthesis



## 1. Introduction

Biodiesel and alcohols are alternative liquid transportation fuels and fuel additives. Alcohols can be synthesized by a number of routes: biomaterial fermentation, alkene hydration, and catalytic conversion of synthesis gas (syngas) [1]. Currently, ZnCu- and ZnCr-oxide catalysts are used at the industrial scale [2]. However, these catalytic systems produce predominantly methanol and small amounts of *i*-butanol, whereas the alcohol mostly used as transportation fuel is ethanol [3]. Vehicles known as flexible fuel vehicles (FFV) can run on pure ethanol, pure gasoline, and any mixture of both. Pure ethanol vehicles are being commercialized in Brazil [4]. A mixture of gasoline and ethanol (E85 has 85% of ethanol) is commercialized in Europe and the USA. Today, most engines can operate with at least 10% ethanol blended into the gasoline (E10) [5]. Many regions around the world allow a mixture of alcohols to be blended with gasoline, in the range from methanol to octanol (boiling point lower than 210 °C). However, due to its toxicity, the methanol content should preferably be low [5]. Another disadvantage of ZnCu- and ZnCr-oxide catalysts is their dramatically low resistance to sulfur poisoning, which is a problem when using sulfur-containing syngas derived from coal and biomass. MoS$_2$-based catalysts are not only resistant to sulfur [6] but even require sulfur to stay in their active sulfidic phase and can produce high amounts of C$_1$-C$_5$ alcohols when modified with potassium. These systems are promising catalysts for the production of alcohols for industrial purposes [6].

Recently, several studies have been published on different aspects of the influence of the composition of the active phase on the mechanism of syngas conversion into alcohols

over supported transition metal sulfides. Using unpromoted $MoS_2$ catalysts produces mainly hydrocarbons, but the promotion of $MoS_2$ by a second transitional metal such as Fe, Co, Ni, and Nb (usually Co) increases alcohols yield at the cost of hydrocarbon selectivity: the promoter atoms work as electron density acceptors on the S-edge of the $MoS_2$ slabs. Furthermore, addition of alkali metals leads to high selectivity to HAS [7,8]: the alkali metals insert between $MoS_2$ crystallites, reduce the metal atoms, and increase the $MoS_2$ slab length and stacking degree [9,10]. In fact, the supports play an important role in the preparation of highly active and selective modified-$CoMoS_2$ catalysts as it impacts electron properties of the formed active phase, morphology, and dispersion. The effect of promoting $MoS_2$ with K, Fe, Co, Ni, and Nb has been studied [9–11]. These promoter atoms suppress hydrogenation reactions and promote the active sites responsible for alcohol synthesis. It is commonly accepted that reactant conversion on these transition metal sulfide (TMS) catalysts proceeds on coordinatively unsaturated sites (CUS) formed on the edges of promoted $MoS_2$ catalysts [9,12,13]. TM atoms participate in the formation of CUS on $MoS_2$ crystallite edges [9,14]. Alkali metal promotion of $MoS_2$ reduces the metal atoms and increases the $MoS_2$ slab length and stacking degree [15,16].

Activated carbons (ACs) have many attractive properties [17,18], such as high stability at high reaction temperatures and pressures [18–20], high surface area, both micro and meso porosity, resistance to acidic and basic conditions, and minimal interaction between support and active phase [21,22]. Activated carbon supports show higher activity for syngas conversion than metal oxide supports ($Al_2O_3$, $SiO_2$, MgO, and $ZrO_2$) because of the weak interaction between the carbon and KCoMoS active phase and the low acidity compared with metal oxides [23,24]. Therefore, we have studied carbon-supported $KCoMoS_2$ catalysts for HAS from syngas. Carbon-coated alumina (CCA) is alumina covered by carbon prepared via pyrolysis of organic material at high temperature in $N_2$. It has excellent mechanical properties [17] and has shown good performance as a support for TMS for alcohol synthesis from syngas. The carbon on the surface of CCA decreases surface hydroxyl groups while maintaining textural properties and reduces interaction between alumina and active phase [20]. The optimal loading of carbon on CCA for this process is 1.7% [25], whereas carbon content above 5% decreases the stacking of $MoS_2$ crystallites on CCA.

There has been an ever-increasing interest in the use of shaped carbon materials with regular structures, including activated carbons, as catalyst support for syngas conversion [18,20]. More than a thousand ACs are available commercially, with different porosities, typically made from different source materials [18] for particular applications.

The aim of the current study is to investigate the effectiveness of carbon materials of different origin used as support for $KCoMoS_2$ catalysts by comparison of structural and catalytic properties of the catalysts supported on various materials ($\gamma$-$Al_2O_3$, CCA, and different types of ACs: AG-3 and BAW) for alcohol production from syngas.

## 2. Results

### 2.1. Characterization

#### 2.1.1. Elemental Analysis and $KCoMoS_2$ Particle Size

XRF elemental analysis data as well as average sulfide slab length and degree of stacking (determined from TEM images) are presented in Table 1. The molybdenum content in the samples is close to 15 wt%. The promotion degree $r$ = Me/(Mo + Me) [molar ratio] ranges from 0.34 to 0.39. The modification degree t = K/(Me + Mo) [molar ratio] ranges from 1.03 to 1.30. Using carbon as a support material increases both the average slab length and the degree of stacking significantly, due to alumina having stronger interaction with the active phase (precursors) [26,27]. Representative TEM images are presented in Figure 1. The slab length and degree of stacking distributions are presented in Figures S1 and S2, respectively.

**Table 1.** Composition of prepared catalysts (XRF) and average slab length and degree of stacking (TEM).

| Catalyst | Content (wt%) | | | Molar Ratio | | Average Slab | |
|---|---|---|---|---|---|---|---|
| | Mo | K | Co | r[1] | t[2] | Length (nm) | Stacking (Layers) |
| $KCoMoS_2/Al_2O_3$ | 15.8 | 10.2 | 5.1 | 0.34 | 1.03 | 6.6 | 1.6 |
| $KCoMoS_2/CCA$ | 14.4 | 11.6 | 4.7 | 0.34 | 1.30 | 11.7 | 3.8 |
| $KCoMoS_2/AG$-3 | 14.9 | 12.3 | 5.7 | 0.39 | 1.24 | 17.0 | 2.8 |
| $KCoMoS_2/BAW$ | 14.8 | 12.0 | 5.2 | 0.36 | 1.27 | 24.5 | 3.2 |

[1] r = Me/(Me + Mo) molar ratio. [2] r = t = K/(Me + Mo) molar ratio.

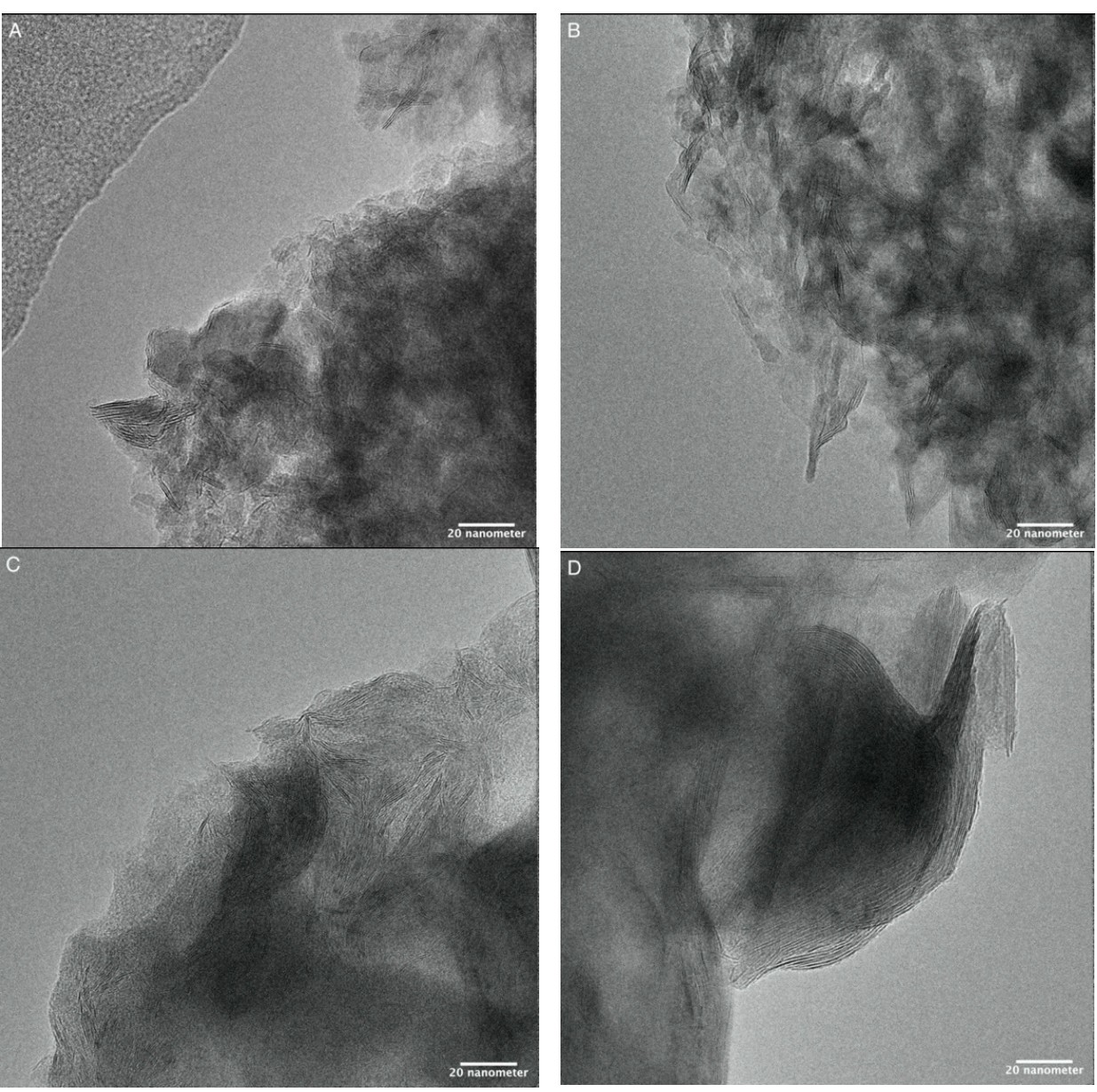

**Figure 1.** Representative TEM images of the four catalysts. (**A**) Cat-Al$_2$O$_3$; (**B**) Cat-CCA; (**C**) Cat-AG-3; (**D**) Cat-BAW.

### 2.1.2. Textural Characteristics

Table 2 shows that the specific surface area (S$_{total}$) of the support materials increases in the order CCA < Al$_2$O$_3$ < BAW < AG-3 and the total pore volume decreases in the order Al$_2$O$_3$ ≥ CCA > AG-3 > BAW. Alumina does not contain micropores at all, and

the specific surface area for the micropores of the other supports increases in the order CCA < BAW < AG-3. The micropore volume increases in the order CCA < AG-3 < BAW. The $Al_2O_3$ and CCA contain practically only mesopores (pore diameter: 4–50 nm), whereas AG-3 and BAW mainly contain micropores. Loading the sulfides onto the support materials leads to a significant decrease in pore surface and volume for all catalysts, indicating the precursor solutions penetrate the pores as expected. The $N_2$ adsorption/desorption isotherms for the support materials and the catalysts are shown in Figure S3.

**Table 2.** Textural characteristics and acidity of supports and catalysts.

| Sample | $S_{total}$, m$^2$/g | $S_{micro}$, m$^2$/g | $S_{meso}$ [1], m$^2$/g | $V_{total}$, cm$^3$/g | $V_{micro}$, cm$^3$/g | $V_{meso}$ [2], cm$^3$/g | Acidity [3], mmol/g |
|---|---|---|---|---|---|---|---|
| $Al_2O_3$ | 161 | 0 | 161 | 0.65 | 0.00 | 0.65 | 286 |
| $KCoMoS_2$/$Al_2O_3$ | 91 | 0 | 91 | 0.29 | 0.00 | 0.29 | 44 |
| CCA | 156 | 13 | 143 | 0.63 | 0.01 | 0.63 | 4 |
| $KCoMoS_2$/CCA | 73 | 0 | 73 | 0.26 | 0.00 | 0.26 | 12 |
| AG-3 | 854 | 753 | 101 | 0.45 | 0.35 | 0.10 | 4 |
| $KCoMoS_2$/AG-3 | 164 | 137 | 27 | 0.09 | 0.06 | 0.03 | 9 |
| BAW | 753 | 642 | 111 | 0.39 | 0.26 | 0.13 | 0 |
| $KCoMoS_2$/BAW | 404 | 365 | 40 | 0.23 | 0.16 | 0.07 | 7 |

[1] $S_{meso} = S_{total} - S_{micro}$. [2] $V_{meso} = V_{total} - V_{micro}$. [3] mmol pyridine adsorbed per gram of support or catalyst.

The alumina-based supports ($Al_2O_3$ and CCA) show $N_2$ uptake limitation at high values of relative pressures ($P/P_0$) and typical curves for type IV adsorption isotherms with H1 type hysteresis [23]. This type of isotherm is typical for mesoporous samples with capillary condensation inside the mesopores. H1 type hysteresis is attributed to materials consisting of agglomerates with narrow pore size distribution.

Adsorption isotherms for the activated carbons (AG-3 and BAW) differ significantly from those for the alumina-based support materials. AG-3 exhibited high uptake even at low relative pressures ($P/P_0$ 0.1–0.2), and a type I isotherm with H4 type hysteresis typical for microporous material. BAW shows a type II isotherm with H4 type hysteresis. The horizontal plateau at high relative pressure values ($P/P_0{\sim}0.9$–1) indicates a large amount of micropores with narrow pore size distribution. H4 type hysteresis is often associated with narrow slit-like pores.

Cat-$Al_2O_3$ and Cat-CCA show type $H_2$ hysteresis. This type of hysteresis is explained by a difference in $N_2$ adsorption and desorption mechanisms that take place in wide pores with a narrow neck ('ink bottle' pores). The observed tails (low pressure hysteresis) in the isotherms recorded for these catalysts indicate either interaction of adsorbate molecules ($N_2$) with the catalyst surface or existence of pores of sizes comparable to the adsorbate molecule size. For activated-carbon-supported catalysts, the presence of the active phase affects neither the isotherm nor the hysteresis type, but it does decrease the surface area and pore volume. Qualitatively, the adsorption isotherms for our activated carbon-based microporous materials and corresponding catalysts, with their typical inflection points, are very similar to those for analogous microporous materials presented in Figures 1 and 2 of [28].

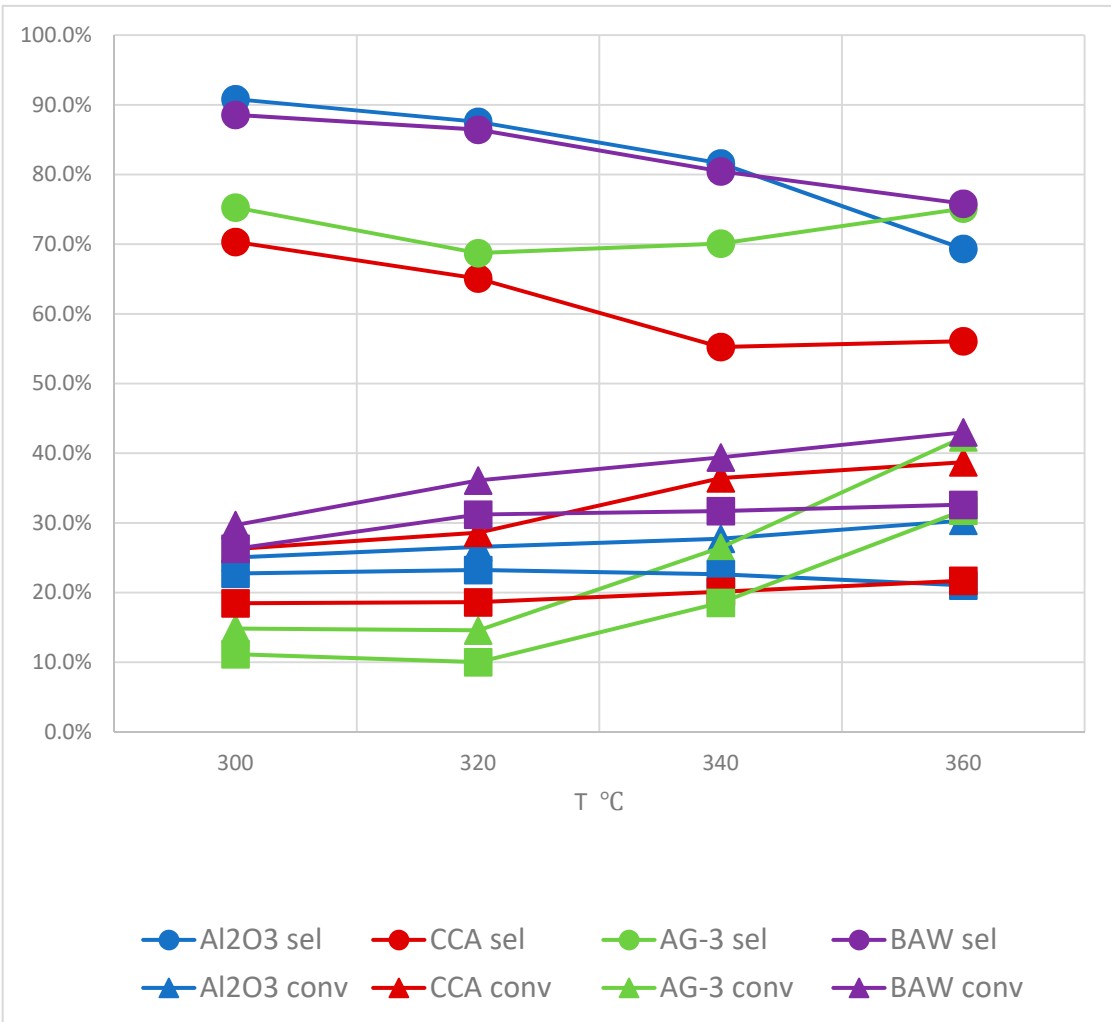

**Figure 2.** Catalytic performance of the KCoMoS$_2$ catalysts on the various support materials. Triangles: CO conversion; Spheres: total alcohol selectivity; Squares: total alcohol yield. Data for the individual alcohols can be found in Table S1.

### 2.1.3. Acidity

The carbon-based support materials do not possess significant acidity but loading with KCoMoS$_2$ creates a small number of the acid sites. Loading alumina with KCoMoS$_2$ blocks most of the acid sites for pyridine adsorption.

### 2.2. Catalytic Experiments

The catalytic data presented in Figure 2 and Table S1 show that as the conversion increases, both total alcohol selectivity and yield indeed increase for the activated-carbon-supported catalysts, whereas they decrease for the alumina-based catalysts. The carbon-supported catalysts show a larger increase in activity with temperature and a lower decrease in total alcohol selectivity. Cat-Al$_2$O$_3$ shows a large increase in selectivity towards the undesired CO$_2$ by-product with increasing reaction temperature, from 9.2% at 300 °C to 24.7% at 360 °C. For Cat-BAW, this increases from 9.1% to only 14.4% over the same temperature range. Actually, our data at 360 °C as presented in Figure 3 show an inverse logarithmic correlation (with R$^2$ = 0.987) between the number of acid sites on the catalysts and the CO conversion. Table 3 summarizes catalytic data from literature as well as those from the current study. It clearly shows the current activities are comparable to or better than those reported earlier, and the current selectivities are superior to those reported earlier.

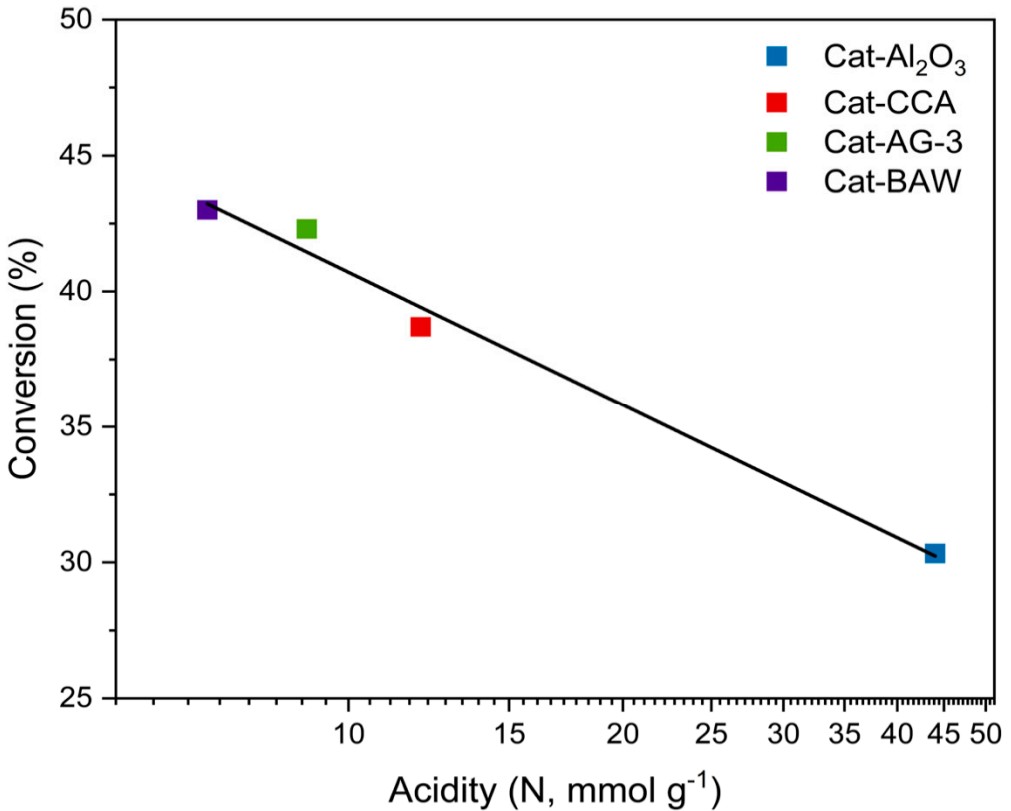

**Figure 3.** Inverse logarithmic relation between the conversion at 360 °C and the number of acid sites on the catalysts as determined from pyridine adsorption.

**Table 3.** Comparison of literature reaction conditions, support nature, and catalytic activity in higher alcohol synthesis from synthesis based on supported K-modified $CoMoS_2$ catalysts.

| Catalyst | GHSV $h^{-1}$ | p MPa | T °C | Conv % | Tot Liq Sel % | Ref |
|---|---|---|---|---|---|---|
| $KCoMoS_2/Al_2O_3$ | 760 | 5 | 340 | 23 | 48 | [29] |
| $KCoMoS_2/CCA$ | 760 | 5 | 340 | 19.2 | 65 | [25] |
| $KCoMoS_2/MWCNT$ | 1200 | 8.3 | 320 | 25 | 40 | [6] |
| $KCoMoS_2/AC\text{-}CGP$ [1] | 1200 | 8.3 | 330 | 44.5 | 27.5 | [30] |
| $KCoMoS_2/AC\text{-}RX3$ [2] | 1200 | 8.3 | 330 | 39.6 | 25.8 | [30] |
| $KCoMoS_2/AC\text{-}Darco$ [3] | 1200 | 8.3 | 330 | 35.6 | 24.8 | [30] |
| $KCoMoS_2/Al_2O_3$ | 760 | 5 | 360 | 30.3 | 69.2 | current [4] |
| $KCoMoS_2/CCA$ | 760 | 5 | 360 | 36.4 | 71.2 | current [4] |
| $KCoMoS_2/AG\text{-}3$ | 760 | 5 | 360 | 42.3 | 75.1 | current [4] |
| $KCoMoS_2/BAW$ | 760 | 5 | 360 | 43.0 | 75.7 | current [4] |

[1] AC-CGP super—Commercial AC (Norit, USA). [2] AC-RX3 extra—Commercial AC (Norit, USA). [3] AC-Darco—Commercial AC (Aldrich, Canada). [4] The current study.

Figure 4 shows the dependence of the chain growth coefficient $\alpha_i$ on a carbon atom number (*i*) in the chain of intermediate products at 360 °C. Differences between the catalysts are small, with Cat-BAW showing a significantly higher $\alpha_4$.

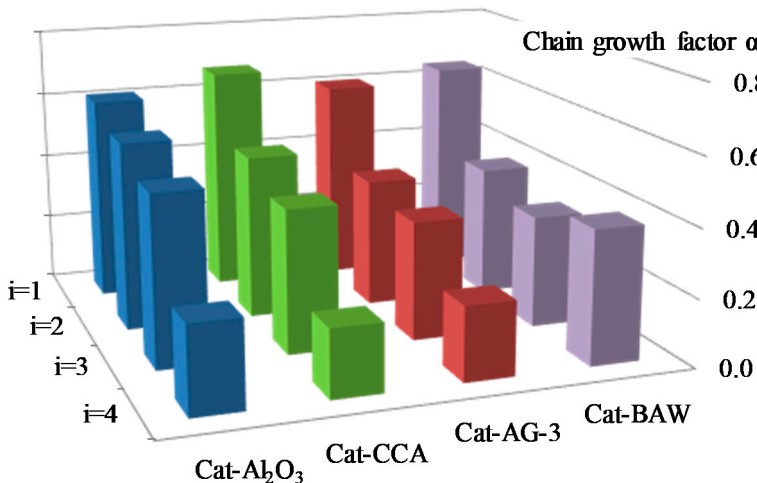

**Figure 4.** Chain growth factors $\alpha_i$ at 360 °C for steps i = 1, 2, 3, 4 over the $KCoMoS_2$ catalysts supported on $Al_2O_3$, CCA, AG-3 and BAW.

## 3. Discussion

The carbon on the CCA decreases the interaction between the active phase and the alumina as well as the hydrogen spillover [25]. Carbon does not coat the surface of alumina uniformly when its content is below 5 wt%, as the organic precursors adsorb preferably on Lewis sites [25,31] and block them. The oxidic precursors for the active phase can only adsorb on the remaining carbon-free alumina surface, leading to sulfide crystallites with a higher stacking number [25].

The reaction network of alcohol formation on $KCoMoS_2$ catalysts has been investigated both experimentally and computationally [9], showing carbon monoxide adsorbs on a surface S vacancy via its carbon atom (C–metal interaction) before being partially hydrogenated, leading to extension of the C=O bond length. As a result, oxygen can coordinate to potassium ions present between $MoS_2$ layers, as shown by IR-spectroscopy [32]. Oxygen coordinated on $K^+$ stabilizes the alkoxyl intermediate and protects the sigma C–O bond from hydrogenolysis to form an alcohol group. An alkyl fragment can react to form an alkane or undergo a chain growth reaction with a new carbon monoxide molecule inserted between the transition metal atom and the adsorbed carbon atom [31].

Strong interaction between active phase and different types of carriers can lead to different electronic states on potassium and cobalt atoms. Acidic sites on the support also influence the catalytic activity. A high content of strong acid sites has been shown to lead to impeding of alcohol formation and accelerating hydrogenation reactions [31,33–36]. As the carbon (modified) supports do not show acidity, they should promote alcohol formation. This is indeed observed.

Next to influencing the selectivity [31,33–36], the acid sites also greatly hamper CO or $H_2$ activation. One possible explanation might be adsorption of the main by-product, water, on the acid sites, blocking access to the active sites via steric hindering. However, it will require a dedicated study to evaluate this hypothesis.

The yield of alcohols seems to increase with the micropore specific surface and volume and decrease with the mesopore specific surface area and volume (Tables 2 and S1)). This contradicts the results reported by Surisetty et al. [6], who found that mesoporous MWCNT as a support gives higher activity than microporous carbon for alkali-modified trimetallic Co–Rh–Mo sulfide catalysts microporous AC. They postulate that the CO conversion and alcohol yield are related to the textural properties of the support such as pore size and mesoporosity rather than its surface area and pore volume. In agreement with our results, they also find that dispersion on mesoporous MWCNT is lower than on microporous AC. This corroborates our findings of large $CoMoS_2$ agglomerates formed on the BAW yielding more alcohols than the higher dispersed slabs on the alumina-based supports (Figure 1 and Tables 1 and S1). Surisetty et al. [6] attribute the higher catalytic activity

of mesoporous MWCNT as a support to decreased diffusion limitations compared to microporous supports. Their results show that even a small amount of mesopores is better than a large amount of micropores. However, our best performing catalysts (Cat-AG-3 and Cat-BAW) contain mainly micropores with just a few mesopores (see Table 2), but also the lowest number of acid sites (see Figure 3). A dedicated study will be required to distinguish between the effects of acidity and porosity, the challenge being to design catalysts with the same acidity but different porosity and vice versa. It would also be a useful challenge to design a method to determine if the number and/or strength of acid sites changes with temperature. The reactants (CO and $H_2$) should not be hindered much by diffusion limitations in micropores, nor should the smaller products. However, the AC-supported catalyst with the largest amount of mesopores, Cat-BAW, shows the highest $\alpha_4$, indicating other factors might be more important. The presence of various functional groups on the carbon surface might be a crucial factor influencing catalytic activity and alcohol selectivity of carbon-supported materials. Such functional groups may strongly interact with the active phase, products, and reactants. The MWCNT support material used in [6] probably contains less functional groups than the activated-carbon-based materials. In this connection, it would be interesting to consider characterizing the surface groups of the carbon support materials in detail.

## 4. Materials and Methods

### 4.1. Preparation of Catalysts

$\gamma$-$Al_2O_3$, CCA, and two different ACs were used as support materials. $\gamma$-$Al_2O_3$ was obtained from Sigma Chemical Co. (St. Louis, MO, USA), crushed and sieved to obtain a particle size fraction of 0.2–0.5 mm. CCA was prepared by impregnating 4g $\gamma$-$Al_2O_3$ with about 15mL of a mixture of glycerol and 2-propanol (1:1) followed by pyrolysis under nitrogen (flow rate 1 L/min) at 200 °C for 40 min and 600 °C for 1 h, using a heating rate of 10 °C/min. The coke content was determined by thermogravimetric analysis (TGA) using a NETZSCH STA 4449 F3 Jupiter apparatus. Thermogravimetric and differential thermogravimetric curves for the CCA were recorded in flowing air from room temperature to 600 °C (heating rate 10 °C/min) [24]. The carbon loading on the CCA support is close to the 1.7% that has been reported to be the optimum loading for alcohol production [25].

Activated carbon AG-3 (commercial trademark АГ-3) was obtained from weakly coking coal crude and coal semi-coke with coal tar pitch binder by preparation of dough, granulation, carbonization, and gas-vapor activation. Activated carbon BAW (commercial trademark БАУ) was manufactured from irregular-shaped charcoal grit via gas-vapor activation at 850–900 °C [26].

The catalyst precursors were prepared by incipient wetness impregnation. Ammonium heptamolybdate tetrahydrate (Alfa Aesar, chemically pure 99%; 5 mmol, 0.48 g) was dissolved in a mixture of 1 mL of $NH_4OH$ (20%) solution and 1.5 mL of distilled water, then mixed with 0.40 g (10 mmol) of KOH (analytical grade, 98%). The produced solution was added to a mixture of cobalt acetate (Alfa Aesar, tetrahydrate, chemically pure 98%; 2.5 mmol) and 1.05 g (5 mmol) of citric acid in 1 mL of distilled water. The impregnated supports (3 g) were dried in flowing air (1 L/min) for 2 h at 60 °C and then for 5 h at 100–110 °C. The catalyst precursors were sulfided in an autoclave using crystalized (elemental) sulfur (1:4 catalyst:sulfur) at 360 °C under $H_2$ at 6.0 MPa for 1 h. The supported catalysts are denoted as Cat-$Al_2O_3$, Cat-CCA, Cat-AG-3 and Cat-BAW, where Cat denotes the active phase—$KCoMoS_2$.

### 4.2. Physical Characterization

#### 4.2.1. Textural Properties

Textural characteristics for supports and catalysts were studied via $N_2$ adsorption and desorption isotherms measured using a Quantachrome Nova 1200e (Anton Paar, Graz, Austria) instrument at 77 K, approximately 0.1 g of each sample, and calibrated sample cells. Oxidic samples were kept under argon flow for 3 h and sulfided samples were

kept under hydrogen flow for 3 h before degassing. The oxidic samples were degassed at 110 °C for 4 h at $10^{-4}$ mm Hg and the sulfided samples at 250 °C for 4 h at $10^{-4}$ mm Hg. The specific surface area was determined using the BET equation. The total pore volume was characterized at a relative pressure $p/p_0 = 0.99$. The mesopore size distribution and volume (considering the adsorption film thickness on the mesopore surface) were calculated from the desorption branch of the isotherm using the Barrett, Joyner, and Halenda (BJH) method [26]. The micropore volume was determined using the t-plot method [27] and by comparing the total pore and mesopore volumes. Volume and pore size values are summarized in Table 2.

A LaB$_6$ Tecnai G2 20F Transmission Electron Microscope (TEM) (ThermoFisher Scientific, Waltham, MA, USA) operated at 200 kV and a FEG Tecnai G2 30F TEM (ThermoFisher Scientific, Waltham, MA, USA) operated at 300 kV were used to characterize the morphology of the sulfided catalysts, dispersed on continuous or Quantifoil™ (Großlöbichau, Germany) R1.2/1.3 microgrid carbon film on copper TEM grids by dry dipping or ethanol suspension. Average slab length and average degree of stacking were evaluated from representative TEM images by manually measuring 300–400 individual slabs using the Fiji software package.

### 4.2.2. Elemental Composition

The elemental composition of the catalysts was determined using an EDX-7000 X-ray fluorescence (XRF) (Shimadzu, Kyoto, Japan) spectrometer with a Rh tube anode operated between 8–200 mA and 15–50 kV. All samples were crushed before measurements. The spectra were processed using the method of fundamental parameters. The elemental composition data are given in Table 1.

### 4.2.3. Acidic Properties

Analysis of acid–base properties of the supported catalysts was performed by determining how much pyridine is adsorbed from a pyridine solution in octane. The pyridine concentration was determined using an SF-103 single-beam scanning UV spectrophotometer (Aquilon, Moscow, Russia). Figure S4 shows the calibration spectra; the adsorption maximum is at 252 nm regardless of the pyridine concentration. Figure S5 shows the calibration line, which yields Equation (1) with R = 0.999.

$$\text{Absorption (D, au)} = 1799.44 \times [\text{pyridine (mol/L)}] \tag{1}$$

The catalysts were stirred in the starting pyridine solution (in octane) for 24 h and removed from the solution before recording the adsorption spectra of the remaining pyridine. Gibbsian adsorption (G, mol/g) was calculated using Equation (2):

$$G = \frac{(C_0 - C_t) \times V}{m} = \frac{(D_0 - D_t) \times V}{m \times \varepsilon \times l} \tag{2}$$

where V is the volume of the solution (10 mL); m is the mass of the sample (0.1 g); $D_0$ and $D_t$ are the optical density at the maximum absorption of pyridine before and after adsorption; l is the thickness of the cuvette (1 cm); $\varepsilon$ is the molar absorption coefficient (extinction, $\varepsilon$ pyridine = $2 \cdot 10^6$ L/(mol·cm), $\varepsilon$ octane = $1 \cdot 10^4$ L/(mol·cm)).

### 4.3. Catalytic Experiments

Syngas conversion was carried out in a fixed-bed flow reactor using $3 \times g$ of sulfided catalyst, P = 5.0 MPa, T = 300–360 °C, mass flow rate 760 L·h$^{-1}$·(g·cat)$^{-1}$, and a feed gas composition of CO:H$_2$:Ar = 45%:45%:10%. The catalysts were evaluated for 4h at each temperature (steps of 20 °C) at T = 300–360 °C. Every 4 h gaseous products were analyzed using a LHM-80 GC with a Thermal Conductivity Detector (TCD) and two one-meter packed columns (molecular sieves CaA (Ar, CH$_4$, CO) and Porapak Q (CO$_2$, C$^{2+}$)). Argon was used as an internal standard for gas chromatography (GC). The liquid

products (alcohols, aldehydes, esters, etc.) were analyzed using a Crystal-2000M GC with a flame ionization detector (FID) and a 50 m HP-FFAP capillary column. Carrier gas was high purity helium for both GCs. The conversion of CO (X) was calculated according to Equation (3) [24]:

$$X = 1 - n_{\text{CO after reaction}} / n_{\text{CO in feed}} \tag{3}$$

The product yield was calculated on a per C atom basis according to Equation (4):

$$Y = N_{\text{C atoms in product}} \times n_{\text{product}} / n_{\text{CO in feed}} \tag{4}$$

The selectivities are summarized in Table S1 and were calculated according to Equation (5):

$$S_i = Y_i / X \tag{5}$$

The carbon chain growth factor $\alpha_i$ was calculated using Equation (6) [31]:

$$\alpha_i = \frac{\sum_{k>i} \frac{Y_k}{k}}{\sum_{k \geq i} \frac{Y_k}{k}} \tag{6}$$

where $\alpha_i$—chain growth factor for the intermediate with i number of carbon atoms; $Y_k$—yield of the component with the k number of carbon atoms. The production of alcohols and hydrocarbons with the same number of carbon atoms was lumped for this calculation.

The factor $\alpha_1$ is the probability of CO insertion to an intermediate containing one carbon atom with the formation of an intermediate with two carbon atoms; $\alpha_2$ that of the next step of CO addition to the intermediate with two carbon atoms, etc.

The chain growth factors determined for reaction at 360 °C are presented in Figure 4 and Table S2.

## 5. Conclusions

Support effects for the synthesis of alcohols from syngas over supported $KCoMoS_2$ catalysts have been studied. $Al_2O_3$, CCA, and two types of commercial activated carbon (AG-3 and BAW) were used as support materials. The catalytic activity increased in the order $Al_2O_3$ < CCA < AG-3 < BAW, inversely correlating with the logarithm of the number of acid sites on the catalysts.

The improved selectivity towards alcohols of the activated-carbon-supported $KCoMoS_2$ seems to be influenced by $KCoMoS_2$ particle size (slab length and degree of stacking; larger particles seem to show improved performance) and porosity (the presence of micropores leading to improved performance), but it will take several dedicated studies to unravel the effects of each parameter in detail.

**Supplementary Materials:** The following are available online at https://www.mdpi.com/article/10.3390/catal11111321/s1, Figure S1: UV absorption spectra of pyridine in octane, Figure S2: Calibration dependence of the UV signal (D) of the solution on the concentration of pyridine (mol/L) at 252 nm, Figure S3: Slab length distribution obtained from minimum 300 individual slabs per sample as recorded from TEM images, Figure S4: Slab degree of stacking distribution obtained from minimum 300 individual slabs per sample as recorded from TEM images, Figure S5. $N_2$ adsorption/desorption isotherms for $Al_2O_3$, CCA, AG-3, and BAW support materials and corresponding $KCoMoS_2$-supported catalysts, Table S1: catalytic selectivities, Table S2: $\alpha$-factors at 360 °C.

**Author Contributions:** Conceptualization, V.M.K.; methodology, V.M.K., M.E.O. and V.V.M.; validation, M.E.O. and V.V.M.; formal analysis, M.E.O., V.V.M., V.S.D., V.M.M. and T.F.S.; investigation, M.E.O. and V.V.M.; resources, V.M.K. and P.J.K.; data curation, M.E.O. and V.V.M.; writing—original draft preparation, M.E.O. and V.V.M.; writing—review and editing, P.J.K.; supervision, V.M.K.; project administration, V.M.K.; funding acquisition, V.M.K. and P.J.K. All authors have read and agreed to the published version of the manuscript.

**Funding:** This research was funded by RFBR, grant number 19-53-60002, and NRF SA, grant number 118919.

**Data Availability Statement:** Original data are available from M.E.O.

**Acknowledgments:** The authors wish to acknowledge the Section of structural studies IOC RAS for electron microscopy analysis, Ekaterina Markova for the pyridine adsorption experiments, and Ziba Rajan for constructing Figure 3.

**Conflicts of Interest:** The authors declare no conflict of interest. The funders had no role in the design of the study; in the collection, analyses, or interpretation of data; in the writing of the manuscript, or in the decision to publish the results.

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
