# Peer review of "Carbon-Supported KCoMoS2 for Alcohol Synthesis from Synthesis Gas"

_catalysts, doi:10.3390/catal11111321_

Round 1

Reviewer 1 Report

The manuscript presents studies on the carbon-supported KCoMoS2 for alcohol synthesis from synthesis gas. In the studies, new materials were characterized by XRF, TEM, BET, acidic properties by UV spectroscopy. In the next step, the catalytic properties of the catalysts supported on various materials for alcohol production from syngas were studied. The manuscript is clear, well-organized, and interesting. Only some parts of the manuscript need improvement, i.e.

  • Authors should clearly explain why decided to used and developed KCoMoS2.
  • The table with a comparison of properties and efficiency of carbon-supported KCoMoS2 with literature data should be added.
  • Fig. S5 – Please change commas to dots.

Author Response

  • Authors should clearly explain why decided to used and developed KCoMoS2. - This explanation with some additional references has been added to the introduction.
  • The table with a comparison of properties and efficiency of carbon-supported KCoMoS2 with literature data should be added. - Table 3 has been added and commented on.
  • Fig. S5 – Please change commas to dots. - This has been changed.

Reviewer 2 Report

In this work, the authors investigated carbon-supported KCoMoS2 for alcohol synthesis from synthesis gas.

I think this paper makes significant contributions to the study of carbon-supported KCoMoS2 for alcohol synthesis from synthesis gas. In my opinion, they found substantial results, and this paper deserves publication.

One of their findings is the inverse logarithmic correlation between the number of acid sites on the catalysts and the CO conversion. They show in figure 3 this correlation at 360° C. Additionally, from figure 2, it is clear that the CO conversion increases with temperature.

In the discussion section, the authors wrote: “Next to influencing the selectivity, the acid sites also greatly hamper CO  or H2 activation. One possible explanation might be adsorption of the main by-product, water, on the acid sites, blocking access to the active sites via steric hindering. However, it will require a dedicated study to evaluate this hypothesis.”

Some discussion on these exciting points seems necessary to me before publication. Is the adsorption of water a function of temperature? Is the number of acidity sites changing with temperature?

Author Response

We thank the reviewer for the  compliments regarding our work. 

We would love to give more discussion on the effect of acidic sites. However, as explained in the manuscript, "A dedicated study will be required to distinguish between the effects of acidity and porosity, the challenge being to design catalysts with the same acidity but different porosity and vice versa." A line has been added, "It would also be a useful challenge to design a method to determine if the number and/or strength of acid sites changes with temperature.", to indicate we are aware of possible temperature effects but not capable at the moment to study those.